# Evidence for single metal two electron oxidative addition and reductive elimination at uranium

Benedict M. Gardner[1], Christos E. Kefalidis [2], Erli Lu[1], Dipti Patel[3], Eric J.L. McInnes [4], Floriana Tuna[4], Ashley J. Wooles[1], Laurent Maron [2] & Stephen T. Liddle [1]

Reversible single-metal two-electron oxidative addition and reductive elimination are common fundamental reactions for transition metals that underpin major catalytic transformations. However, these reactions have never been observed together in the f-block because these metals exhibit irreversible one- or multi-electron oxidation or reduction reactions. Here we report that azobenzene oxidises sterically and electronically unsaturated uranium(III) complexes to afford a uranium(V)-imido complex in a reaction that satisfies all criteria of a single-metal two-electron oxidative addition. Thermolysis of this complex promotes extrusion of azobenzene, where H-/D-isotopic labelling finds no isotopomer cross-over and the non-reactivity of a nitrene-trap suggests that nitrenes are not generated and thus a reductive elimination has occurred. Though not optimally balanced in this case, this work presents evidence that classical d-block redox chemistry can be performed reversibly by f-block metals, and that uranium can thus mimic elementary transition metal reactivity, which may lead to the discovery of new f-block catalysis.

[1] School of Chemistry, The University of Manchester, Oxford Road, Manchester M13 9PL, UK. [2] LPCNO, CNRS & INSA, Université Paul Sabatier, 135 Avenue de Rangueil, Toulouse 31077, France. [3] School of Chemistry, University of Nottingham, University Park, Nottingham NG7 2RD, UK. [4] EPSRC National UK EPR Facility, School of Chemistry and Photon Science Institute, The University of Manchester, Oxford Road, Manchester M13 9PL, UK. Benedict M. Gardner, Christos E. Kefalidis, Erli Lu and Dipti Patel contributed equally to this work. Correspondence and requests for materials should be addressed to L.M. (email: laurent.maron@irsamc.ups-tlse.fr) or to S.T.L. (email: steve.liddle@manchester.ac.uk)

Redox chemistry is a defining feature of transition metal (d-block) chemistry. Within this realm, oxidative addition, first discovered over 50 years ago, is a fundamentally important and elementary transformation[1]. Two types of oxidative addition are known[2,3], involving either two single-electron oxidations, involving two metal centres ($ML_n$, $L_n$ = ancillary ligands) or a binuclear complex (Fig. 1a), or the more classical single-metal two-electron reaction (Fig. 1b). In order to define these reactions, a number of criteria are applied to classify them, where for the former oxidative addition reaction of type (a), the oxidation state (OS), valence electron (VE) and coordination number (CN) all increase by one, the d-electron count ($d^n$) decreases by one, and new covalent M–A and M–B bonds are made at different metal centres. In the latter oxidative addition reaction of type (b), the O.S., V.E., and C.N. all increase by two, the $d^n$ count becomes $d^{n-2}$, and two new covalent M–A and M–B bonds are made at the same metal centre. Of the two types of oxidative addition reaction, which describe the overall reaction with no mechanistic implications, the latter is the most important type, and the reverse reaction is defined as reductive elimination; together these two principal reaction types constitute key steps that underpin most catalytic reactions[4]. With notable main group exceptions such as Grignard (and heavier group 2 congeners) formation (e.g., $Mg^0 + RX \rightarrow RMg^{II}X$)[5–7] and oxidative additions/reductive eliminations involving group 13–15 elements[8–20], this mode of reactivity generally remains the preserve of transition metals.

In contrast, the physicochemical properties of the f-block metals render them generally unable to support classical oxidative addition and reductive elimination reactions because the lanthanides and actinides cannot typically access two electron metal-based redox couples, though irreversible Grignard type reactions (e.g., $M^0 + RI \rightarrow RM^{II}I$; M = Eu, Yb, Sm, R = Me, Et, Ph)[21], which are oxidative additions overall, are known; their reactivity is instead usually defined by single-electron transfers and σ-bond metathesis chemistries that exploit their highly electropositive and polarising natures. Indeed, f-block catalysts can be highly active in σ-bond metathesis reactions[22], but despite decades of f-block research there are no examples of any lanthanide or actinide complexes that perform pure, classical oxidative addition and and reductive elimination reactions. It should be noted that in recent years some spectacular uranium-mediated multi-electron transfer reactions have been reported[23,24], but these utilise metal-ligand redox cooperativity and even when an oxidative addition or reductive elimination is observed it is irreversible and/or does not fit the above definitions[25,26]. If reversible oxidative addition and reductive eliminations could be established for any f-block complex, this would demonstrate transition metal-like reactivity and that these elements might be harnessed in new types of catalysis[27].

When contemplating introducing classical oxidative addition and reductive elimination reactions to f-block chemistry, a number of factors need to be addressed. Heavier elements are more likely to react since their VEs are less tightly bound than in lighter elements. An electron rich, low OS metal will be more oxidisable. Hard σ-donor ligands will favour oxidative addition since they stabilise the resulting higher OS of the metal. Relatively small, sterically undemanding ligands and a large metal will favour oxidative addition, as the coordination sphere of the metal will not be overcrowded. Strong M–A and M–B bonds and a weak A–B bond will favour oxidative addition, but those bond energetics are often finely balanced, resulting in oxidative addition and reductive elimination being viewed as a reversible process, Fig. 1b. With these considerations noted, we concluded that uranium, well known to exhibit variable OSs, represents a promising f-block metal with which to target oxidative addition and

**Fig. 1** Principal types of oxidative addition reaction observed with d-block metals. **a** Two single-electron oxidations of an A–B bond of a substrate at either two transition metal centres (M) or a binuclear complex resulting in changes of +1 to the oxidation state (OS), valence electron (VE) and coordination number (CN) of the metals and a reduction of d-electrons by one. **b** Classical two-electron oxidative addition, the reverse of which is reductive elimination, of an A–B bond with a single transition metal centre, resulting in changes of +2 to the metal OS, VE, CN, and a reduction of d-electrons by two

reductive elimination since its properties compare favourably with the above criteria. However, although examples of oxidative addition-type behaviour of substrates by uranium, which are distinct to two-electron oxidations of uranium to give terminal mono-oxo and -imido ligands[28,29], are known or proposed[25], they are limited to examples that do not conform to the classical definition. For example, cooperative multi-metallic redox transformations (Type (a) in Fig. 1) utilising multiple single-electron uranium redox couples [U(III) to U(IV) or U(V) to U(VI)], where one new covalent uranium-ligand bond per uranium centre is formed are known[30–36]. Non-innocent ligands can provide multi-electron reservoirs with apparent oxidation and one or two new uranium-ligand covalent bonds are formed but the formal uranium OS is unchanged in reactant and product[37–39], or a combination of uranium and non-innocent ligand redox reactions can occur[40–43]. The electron paramagnetic resonance (EPR) data in one study suggest possible oxidative addition of water to uranium(III)[44]. Where reductive elimination is concerned, few examples exist. For instance, biphenyl is known to eliminate from $[UO_2Ph_2]$ to give $UO_2$[45]. Elimination of oxidatively coupled bibenzyl from tetrabenzyl uranium following addition of a non-innocent diazabutadiene (DAB) ligand has been reported, but the formal OS of uranium remains (IV) in the reactant and product and the benzyl electrons reduce the DAB ligand and not uranium[46]. H-H and C–H formal reductive eliminations can generate a masked form of '$[U(C_5Me_5)_2]$', but the formal OS of uranium remains (IV) in reactant and product[47]. Lastly, bimetallic reductive elimination of dihydrogen from uranium hydrides, exploiting multiple single-electron U(IV) to U(III) redox couples, is known [reverse of Type (a) in Fig. 1][40,48]. Thus, noting catalytic reduction of azides[49] and reversible bimetallic one-electron, per uranium(III) ion, addition-elimination reactions of pyrazine[50] that do not fit the definitions above, an f-block system exhibiting reversible classical oxidative addition and reductive elimination is yet to be realised.

Here, we report evidence for bona fide oxidative addition and reductive elimination reaction at an f-block centre. We exploited a reactive, sterically open and electronically unsaturated uranium (III) triamide complex that supports a reversible two-electron metal-centred U(III)-U(V) redox couple. This oxidative addition-reductive elimination couple is not well-balanced, but it suggests that the idea that f-block elements can support such reactivity is valid and could form the basis for new catalytic cycles supported

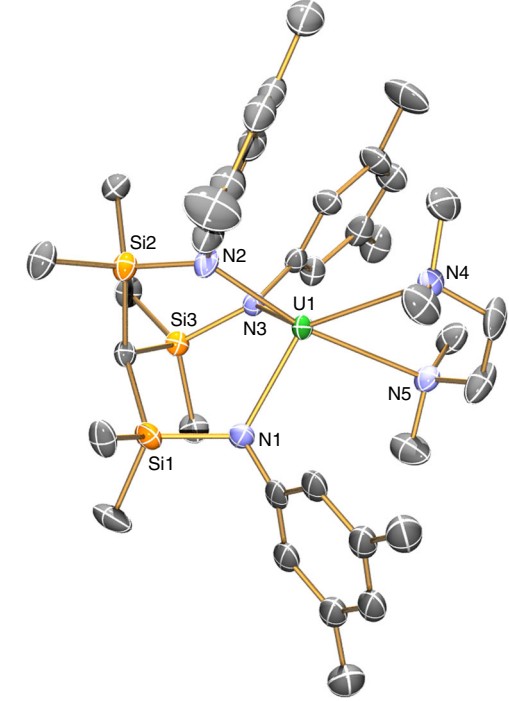

**Fig. 2** Synthesis and reactivity of **3**. Reduction of **1** by different routes involving either reduction in the presence of stabilising polyamines to give **2**.tmeda or **2**.pmdeta, which react with azobenzene or D$_{10}$-labelled azobenzene to give **3**, or a one-pot approach via **2**. Thermolysis of **3** extrudes azobenzenes by a concerted process to presumably regenerate **2**, but the latter decomposes under thermolytic conditions. Ar = 3,5-Me$_2$C$_6$H$_3$

by elementary reactions that are usually restricted to transition metals.

## Results

**Precursor synthesis and the oxidative addition product.** We previously reported that reduction of the uranium(IV) triamide complex [U(Ts$^{Xy}$)(Cl)(THF)] [**1**, Ts$^{Xy}$ = HC(SiMe$_2$NAr)$_3$, Ar = 3,5-Me$_2$C$_6$H$_3$] with potassium graphite in the presence of toluene afforded the formal diuranium(V) arene inverted sandwich complex [{U(Ts$^{Xy}$)}$_2$($\mu$-$\eta^6$:$\eta^6$-C$_6$H$_5$Me)][51], but in hexane solvent the putative uranium(III) complex [U(Ts$^{Xy}$)] (**2**) that is generated by reduction of **1** activates one of the N-aryl bonds of the Ts$^{Xy}$ ligand to generate the dinuclear imido-aryl-bridged complex [U{HC(SiMe$_2$NAr)$_2$(SiMe$_2$-$\mu$-N)}($\mu$-$\eta^1$:$\eta^1$-Ar)U(Ts$^{Xy}$)][51]. This ligand-cannibalisation reactivity, which is symptomatic of reactive low valent uranium(III), suggests that **2** is very reactive due to coordinative and electronic unsaturation and thus might be capable of effecting oxidative addition of a substrate. Because of the high reactivity of **2**, it must be generated in situ and then rapidly reacted on which has prevented us from characterising it. However, in order to better understand the nature of **2**, we utilised the neutral, multi-dentate co-ligands Me$_2$NCH$_2$CH$_2$NMe$_2$ (tmeda) and MeN(CH$_2$CH$_2$NMe$_2$)$_2$ (pmdeta) (See Supplementary Methods) to prepare and isolate the two uranium(III) adduct complexes [U(Ts$^{Xy}$)(L)] (L = tmeda, **2**.tmeda; L = pmdeta, **2**.pmdeta). This strategy is successful since N-aryl cleavage reactions are completely supressed and **2**.tmeda and **2**.pmdeta, which are highly soluble complexes, can be isolated as exceedingly air- and moisture-sensitive, dark violet crystalline solids in 33 and 56% yield (Fig. 2), respectively.

Treatment of **2**.tmeda or **2**.pmdeta with half a molar equivalent of azobenzene (PhN=NPh) in hexanes afforded, after work-up and recrystallisation, brown blocks of the uranium(V)-imido dimer oxidative addition product [{U(Ts$^{Xy}$)($\mu$-NPh)}$_2$] (**3**), typically in 47% isolated crystalline yield (Fig. 2); inspection of the mother liquor by $^1$H nuclear magnetic resonance (NMR)

**Fig. 3** Molecular structure of [U(Ts$^{Xy}$)(tmeda)] (**2**.tmeda) at 120 K with 40% probability ellipsoids. Hydrogen atoms and minor disorder components are omitted for clarity. Selected distances: **2**.tmeda−U1-N1 2.307(8), U1-N2 2.310(9), U1-N3 2.320(7), U1-N4 2.771(8), U1-N4A 2.760 (7) Å

spectroscopy, and comparison to those of **2**.tmeda and **2**.pmdeta (See Supplementary Figs. 1-4) suggests that the reaction is essentially quantitative in nature and the solubility of **3** dictates the crystalline yield. Encouraged by this result, we also find that preparing **2** in situ in the presence of azobenzene also affords

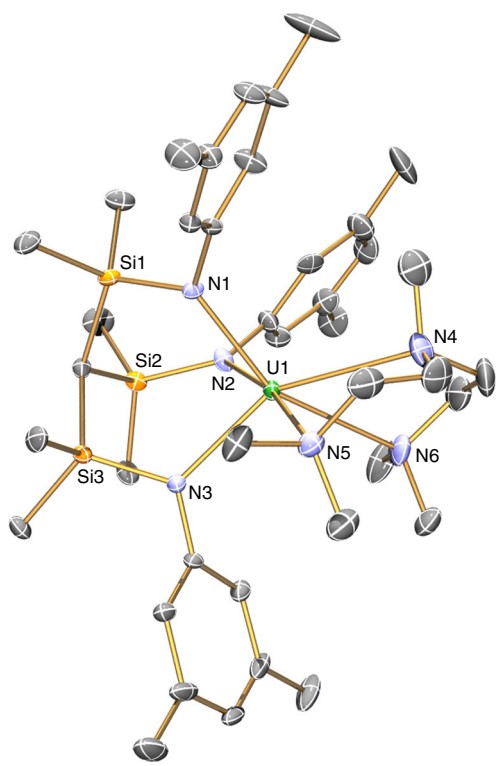

**Fig. 4** Molecular structure of [U(Ts$^{Xy}$)(pmdeta)] (**2**.pmdeta) at 120 K with 40% probability ellipsoids. Hydrogen atoms and minor disorder components are omitted for clarity. Selected distances: **2**.pmdeta - U1-N1 2.373(5), U1-N2 2.394(6), U1-N3 2.355(5), U1-N4 2.831(7), U1-N5 2.866 (7), U1-N6 2.899(6) Å

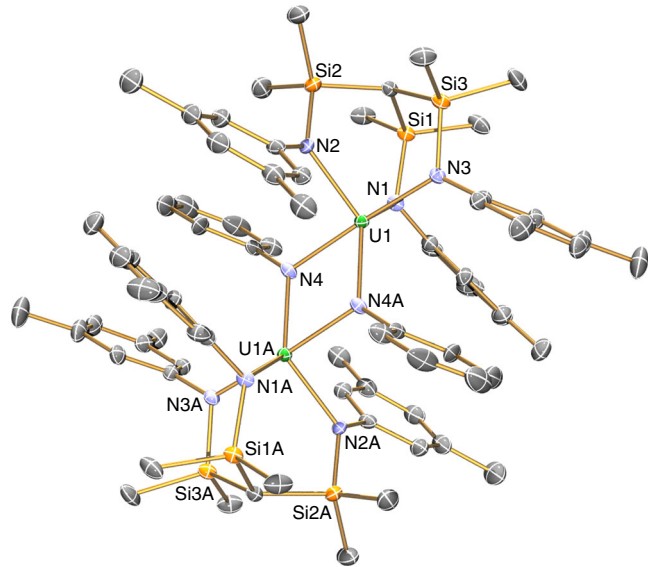

**Fig. 5** Molecular structure of [{U(Ts$^{Xy}$)(μ–NPh)}$_2$] (**3**) at 90 K with 40% probability ellipsoids. Hydrogen atoms and minor disorder components are omitted for clarity. **3**-D$_{10}$ is isostructural to **3**. Selected distances: **3**-U1-N1 2.221(4), U1-N2 2.228(4), U1-N3 2.205(4), U1-N4 2.208(4), U1-N4A 2.210 (4) Å

**3** in 47% yield (Fig. 2). In order to confirm the synthesis of **3**, we also prepared it independently from the aforementioned arene complex [{U(Ts$^{Xy}$)}$_2$(μ-η$^6$:η$^6$-C$_6$H$_5$Me)][51] with concomitant elimination of toluene, but in reduced 29% crystalline yield. This latter reaction is reminiscent of the reaction of the diuranium arene inverted sandwich complex [K$_2$I][{U(NCMes-Bu$^t$)$_3$}$_2$(μ-η$^6$:η$^6$-C$_6$H$_5$Me)] (Mes = 2,4,6-Me$_3$C$_6$H$_2$) with azobenzene which afforded the uranium(V)-imido dimer [{U (NCMesBu$^t$)$_3$(μ–NPh)}$_2$][52].

**Solid state structures.** The solid-state structures of **2**.tmeda and **2**.pmdeta were determined by single crystal X-ray diffraction and are illustrated in Figs. 3 and 4 with selected bond lengths (See Supplementary Tables 1 and 2). The uranium(III) centres are coordinated to the tridentate Ts$^{Xy}$ ligand through the three amide donor atoms leaving the remaining coordination hemispheres to be completed by the bi- and tridentate polyamine ligands. The striking feature of the structures of **2**.tmeda and **2**.pmdeta is that if they are considered without the stabilising amine then the coordination sphere of the uranium(III) ion would clearly be extraordinarily exposed since the Ts$^{Xy}$ ligand barely occupies a hemisphere of coordination space at uranium, which nicely accounts for the high reactivity of **2**. The U–N$_{amide}$ distances span the range 2.307(8)–2.394(6) Å for **2**.tmeda and **2**.pmdeta, which considering their five- and six-coordinate uranium(III) ions compares very well to the U–N$_{amide}$ distance of 2.320(4) Å in three-coordinate [U{N(SiMe$_3$)$_2$}$_3$];[53] this is consistent with the uranium(III) formulations of **2**.tmeda and **2**.pmdeta, since uranium(IV) and (V)-amide distances tend to be shorter at ~2.2 Å. For example, the U-N$_{amide}$ distances in [{U(Ts$^{Xy}$)}$_2$(μ-η$^6$:η$^6$-C$_6$H$_5$Me)] span the range 2.212(3)–2.239(3) Å.[51] The U–N$_{amine}$ distances span the range 2.760(7)–2.900(6) Å and are

unexceptional, though the U-N$_{amine}$ distances are notably shorter for **2**.tmeda than **2**.pmdeta in-line with the respective uranium CNs of those complexes[54].

The molecular structure of **3** was confirmed by single crystal X-ray diffraction and is illustrated in Fig. 5 (for the isostructural D$_{10}$-analogue prepared using D$_{10}$–PhNNPh see Supplementary Fig. 5). The salient feature of **3** is its dimeric centrosymmetric formulation with bridging imido groups to give five-coordinate uranium centres. The U–N$_{amide}$ distances in **3** span the range 2.205(4)–2.228(4) Å, which is ~0.15 Å shorter than the corresponding distances in **2**.tmeda and **2**.pmdeta, and this range compares well to the U–N distances in pentavalent [{U (Ts$^{Xy}$)}$_2$(μ-η$^6$:η$^6$-C$_6$H$_5$Me)][51] and [{U(NCMesBu$^t$)$_3$(μ–NPh)}$_2$][52] which is consistent with a uranium(V) formulation. The bridging U–N$_{imido}$ distances of 2.208(4) and 2.210(4) Å are essentially indistinguishable from the U–N$_{amide}$ distances reflecting their bridging nature; for comparison, uranium(V) terminal imido bond lengths tend to be ~1.95 Å[55]. Both imido phenyl rings are orientated perpendicular to the uranium–uranium vector, so neither of the imido centres can be considered to be doubly-bonded to one uranium and datively-bound to the other uranium centre, which is consistent with the symmetrical nature of the U$_2$N$_2$ four-membered ring.

**Characterisation data.** In order to probe the formal OSs of uranium in **2**.tmeda, **2**.pmdeta, and **3**, and hence unambiguously confirm the occurrence of classical oxidative addition, we examined their ultraviolet/visible/near-infrared (UV/Vis/NIR) electronic absorption and EPR spectra, and magnetic properties (Supplementary Figs. 6–14). The electronic absorption spectrum of **3** in toluene exhibits broad absorptions at 6570, 7650, and 9815 cm$^{-1}$ ($\varepsilon$ = 40–70 M$^{-1}$ cm$^{-1}$) in the NIR region, which are characteristic of Laporte forbidden 5$f$ → 5$f$ transitions for uranium(V) from the $^2F_{5/2}$ ground state to the $^2F_{7/2}$ excited state electronic manifold[56], and a strong ligand to metal charge transfer (LMCT) band which tails in from the UV-region to ~10,000 cm$^{-1}$. The electronic absorption spectrum of **2**.tmeda and **2**.pmdeta in toluene are distinct from that of **3**, but also exhibit Laporte forbidden 5$f$ → 5$f$ transitions in the NIR region

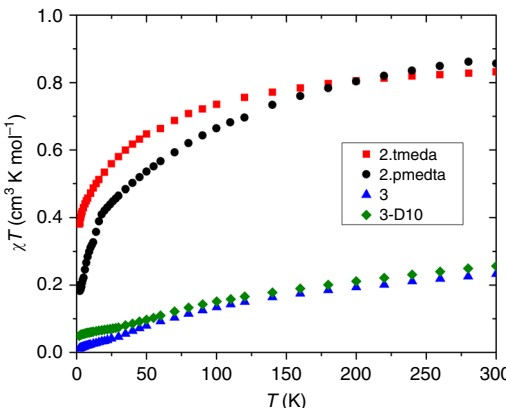

**Fig. 6** Temperature-dependent magnetic susceptibility ($\chi$) data as $\chi T(T)$ for the compounds in this study. Trivalent **2**.tmeda = black circles; Trivalent **2**.pmedta = red squares; Pentavalent dimeric **3** = green triangles; Pentavalent **3**-D$_{10}$ = blue triangles. The data were measured in an applied magnetic field of 0.5 kG

($\varepsilon = 60$–$80$ M$^{-1}$ cm$^{-1}$). However, the characteristic Laporte allowed $5f \rightarrow 6d$ transitions for uranium(III)[57–59], which are usually observed around 17,000 cm$^{-1}$ are observable for **2**.tmeda supporting the OS assignment, though for **2**.pmedta those absorptions are obscured by a strong LMCT band that extends from well into the UV-region. Variable temperature superconducting quantum interference device (SQUID) magnetometry on powdered **2**.tmeda and **2**.pmedta (Fig. 6) reveals $\chi T$ products of ~0.85 cm$^3$ K mol$^{-1}$ at 298 K (corresponding to an effective magnetic moment of ~2.6 $\mu_B$; $\chi$ = molar magnetic susceptibility, $T$ is the temperature). $\chi T$ decreases on cooling, and AC susceptibility studies give low temperature plateaus in $\chi'T$ (where $\chi'$ is the in-phase component), of 0.3–0.4 cm$^3$ K mol$^{-1}$, which is consistent with the lowest energy magnetic Kramers doublets. EPR spectra of powdered **2**.tmeda and **2**.pmedta at 5 K give highly anisotropic effective $g$-values; those of **2**.pmedta ($g_{eff} = 4.0$, 1.6 and 0.7) are similar to those of trivalent [U{N(CH$_2$CH$_2$NSi-Pr$^i_3$)$_3$}][60] (for **2**.tmeda only the highest $g$-value of ~4.2 is clearly resolved). Taken together, these data are consistent with the formal uranium(III) OS. In contrast, $\chi T$ for **3** is 0.23 cm$^3$ K mol$^{-1}$ per uranium ion (1.36 $\mu_B$ per U ion) at 298 K, and decreases steadily towards nil on cooling (Fig. 6). In the high temperature (50–300 K) regime, the magnetic data of **3** show Curie–Weiss behaviour with a Curie constant of 0.37 cm$^3$ K mol$^{-1}$ (1.73 $\mu_B$) per uranium ion. These magnetic data are consistent with uranium (V) with antiferromagnetic coupling between the metal ions, and lie in the range for well-characterised and structurally related uranium(V) dimers[61]. A diamagnetic ground state for dimeric **3** is confirmed by a low temperature magnetisation of <0.1 $\mu_B$ at 2 K and 7 T and the lack of an EPR spectrum.

**Reductive elimination studies**. While initially attempting to isolate pure crystalline **3**, we noticed that when we placed **3** under dynamic vacuum and gentle heat to remove residual solvent from washing during work-up an orange material slowly began to extrude from **3**. The rate of extrusion can be moderately increased by heating **3** to >100 °C under sublimation conditions, but it is kinetically hindered by the crystalline nature of isolated **3** even when finely-ground. Collection of the orange material and analysis by NMR spectroscopy revealed it to be azobenzene, which was confirmed by comparison of its NMR spectra to those of an authentic sample from a commercial supplier and by a peak at $m/z = 181$ ({PhNNPh–H}$^-$) in the negative mode electrospray ionisation mass spectrum of this material (Supplementary

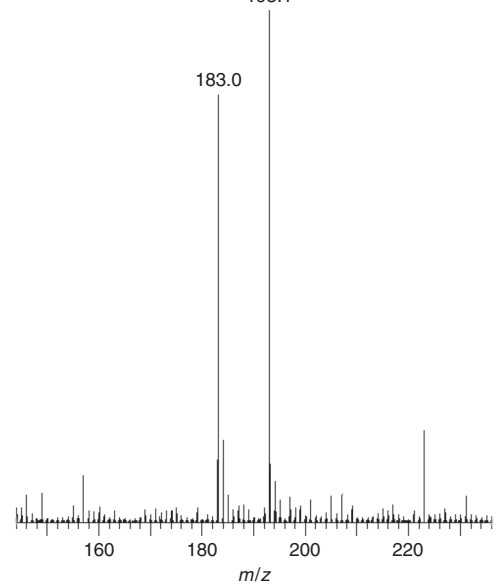

**Fig. 7** Positive-ion mode electrospray ionisation mass spectrometry of the azobenzene product obtained from the reductive elimination of **3**. The signals at $m/z$ 183.0 and 193.1 confirm the presence of exclusively H$_{10}$–PhNNPh and D$_{10}$–PhNNPh, respectively, with no H$_5$/D$_5$–PhNNPh even though the reductive elimination is conducted under a thermal regime

Figs. 15–19). In order to confirm this result, we isolated **3** by washing the crystalline material with dry pentane and drying under a nitrogen flow, then took this material and heated it in a sublimation tube, after confirming purity by NMR spectroscopy and elemental analysis, with an identical result.

Uranium(III) is strongly reducing and uranium(V) is strongly oxidising, and it would appear that **3** is close enough to the cusp of this redox couple so that the initial oxidative addition reaction that produces **3** can be reversed by reductive elimination when thermally instigated. This view is supported by the fact that treatment of **3** with sources of H$^+$, e.g., water, results in decomposition and the liberation of PhNH$_2$, as assayed by $^1$H NMR spectroscopy, with no PhNNPh detected under those conditions.

The extrusion of PhNNPh from **3** suggests that a concerted reductive elimination is occurring, but monometallic and/or nitrene mechanisms would compromise the claim of reductive elimination from **3**. Therefore, we prepared **3**-D$_{10}$ using D$_{10}$–PhNNPh and thermolysed a homogenous 50:50 mixture of **3** and **3**-D$_{10}$; if a concerted reductive elimination mechanism operates pure H$_{10}$–PhNNPh and D$_{10}$-PhNNPh would be obtained but if monometallic intermediates or nitrenes are generated then H$_5$/D$_5$–PhNNPh would be formed as well as H$_{10}$–PhNNPh and D$_{10}$–PhNNPh. Experimentally, we find that only H$_{10}$–PhNNPh and D$_{10}$–PhNNPh are formed (Supplementary Figs. 20 and 21), as evidenced by electrospray ionisation mass spectrometry (Supplementary Fig. 22), which shows peaks at $m/z$ 183 ({PhNNPhH}$^+$) and 193 ({D$_{10}$–PhNNPhH}$^+$) in positive ion mode, but the $m/z$ 188 peak for ({H$_5$/D$_5$–PhNNPhH}$^+$) is absent (Fig. 7). Further, when Ph$_2$C=CPh$_2$, an established nitrene trapping agent ($Z$- or $E$-Ph(H)C=C(H)Ph are too volatile), is mixed into the reductive elimination mixture no aziridene products that would be expected from nitrene generation are observed, and only PhNNPh is isolated again.

**Mechanistic studies**. Oxidative addition describes the overall reaction and has no mechanistic implications. However, to be

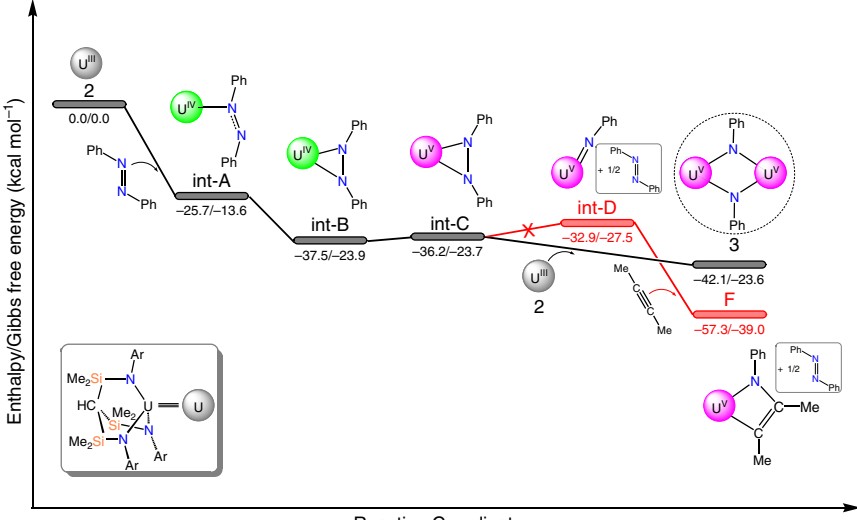

**Fig. 8** Computed reaction profile of the reaction of **2** with PhNNPh to give **3** via an overall oxidative addition reaction. Values given are quoted as computed enthalpy/Gibbs free energies

considered a true oxidative addition the reactions of **2**.tmeda and **2**.pmdeta should not proceed via a terminal uranium-imide monomer. In order to probe this aspect, since in situ probing of this reaction by NMR or optical spectroscopies were not practicable, we modelled the reaction computationally since DFT reaction profile calculations have proven their ability to reliably describe the redox activity of f-element molecules; we provide ΔH and ΔG data, and note that the latter presents essentially the same picture as the former, but use the former in our discussions since the latter introduces errors from the way ΔS is calculated within the harmonic approximation (See Supplementary Tables 3–15)[62].

At the B3PW91 level of theory, we examined the formal four-electron reduction of PhNNPh in the presence of **2**, with the polyamine ligands omitted from the calculations for computational efficacy (Fig. 8). Overall, the reaction of two equivalents of **2** with PhNNPh to give **3** is found to be highly favoured enthalpically ($-42.1$ kcal mol$^{-1}$ overall), where the complete cleavage and reduction of the PhNNPh is readily apparent along with oxidation of each uranium from +3 to +5 OSs as evidenced by excellent agreement of key metrical bond length data (Supplementary Fig. 23). Initially, one electron reduction of azobenzene, induced by coordination to uranium is found to be exothermic by 25.7 kcal mol$^{-1}$. The formal OS of the uranium ion in this species is IV, which is apparent from the 0.08 Å elongation of the N–N distance of the azobenzene with respect to the computed distance of 1.257 Å for free azobenzene in the gas-phase, and we note that N=N distances span the range ~1.10–1.25 Å in crystallographically authenticated examples of free-azobenzene[54]. The elongated N–N distance is within the range of experimentally determined mono-reduced azobenzenes in f-element chemistry[63–66], and the spin density is also commensurate with the uranium(IV) assignment (Supplementary Fig. 24). It should be noted that the coordinated azobenzene radical is now somewhat distorted with respect to the free molecule, but still maintains its *trans* configuration. Isomerisation of the coordinated azobenzene to obtain a *cis* conformer affords an energetically more stable intermediate, **int-B**, being $-37.5$ kcal mol$^{-1}$ with respect to **2**. Surprisingly, examination of the spin density distribution reveals a broken symmetry state, with two α-spin electrons located on the uranium ion and one β-spin diffused onto the azobenzene fragment. Interestingly, an intermediate with minor geometry variations that is extremely close in energy could be located, which corresponds to the intermediate **int-C**. In the

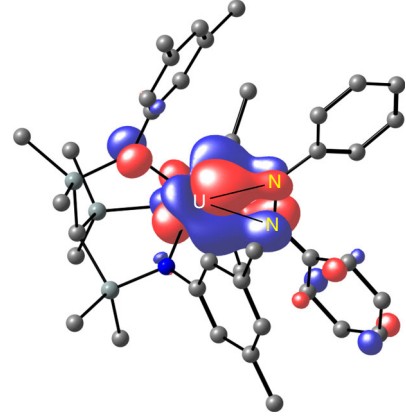

**Fig. 9** SOMO orbital of the [U(Ts$^{Xy}$)(κ$^2$-PhNNPh)] intermediate **int-C**. This shows the δ-type uranium-azobenzene bonding interaction involving a π$^\star$ azobenzene orbital interacting with a uranium 5f orbital of general setting $m_l = 2$ type parentage. Hydrogen atoms are omitted for clarity

latter, the N–N distance is 1.41 Å, which is in the region of doubly reduced azobenzenes (See Supplementary Fig. 25)[64–66]. Hence, these two distinct intermediates are very close in energy, and can be viewed as the two limiting forms of the intermediate that would have strong multi-reference character and we note that this presents a similar spin density picture to that found in ytterbium intermediate-valence compounds[67]. Inspection of the molecular orbitals of **int-C** reveals an intriguing bonding situation; specifically, its singly occupied molecular orbital principally represents overlap between a general setting $m_l = 2$ type 5f orbital with the N–N π$^\star$-orbital of the azobenzene (Fig. 9). Fascinatingly, four lobes from the 5f orbital overlap with the four lobes of the aforementioned π$^\star$-orbital in a δ-type bonding motif. It should be noted that the same bonding situation is found in **int-B**.

The formation of a terminal uranium(V)-imido monomer complex, **int-D**, was investigated. This reaction is slightly endothermic compared to **int-B** and **int-C** (4.6 and 3.3 kcal mol$^{-1}$, respectively) so that these two complexes could in principle be in equilibrium. However, the coordination of a second molecule of **2** to [U(Ts$^{Xy}$)(κ$^2$-PhNNPh)] and subsequent two-electron reduction, yielding the final bis-imido bimetallic complex **3**, is

exothermic by a further 5.9 kcal mol$^{-1}$. On the basis of those data it is difficult to ensure that the overall four-electron reduction is direct and not involving formation of two terminal imido complexes by two two-electron reductions that subsequently dimerise. However, given the sterically wide-open coordination sphere of uranium with a Ts$^{Xy}$-ligand set a terminal imido complex would certainly react with unsaturated substrates. Therefore, we examined reactions of **2.tmeda** and **2.pmdeta** with PhNNPh in the presence of alkynes and also the addition of alkynes to already prepared and isolated **3**. If a terminal imido complex were ever to exist as an intermediate, then it would undergo a [2 + 2]-cycloaddition to yield a metallacyclobutadiene-type complex formation. However, a range of alkynes (MeC≡CMe, PhC≡CPh, Bu$^t$C≡CH, Me$_3$SiC≡CH) are found experimentally to not react, and even the polar and thus reactive Bu$^t$C≡P does not react where it has previously been found to be much more reactive than PhC≡CPh[68–71]. These observations, however, are in agreement with the calculated reaction profile (see Supplementary Fig. 26), which for MeC≡CMe as an exemplar reveals that the hypothetical product of a [2 + 2]-cycloaddition between [U(Ts$^{Xy}$)(NPh)] and MeC≡CMe exhibits an activation barrier of 12.3 kcal mol$^{-1}$ uphill and is thus disfavoured though in principle is accessible under experimental conditions. We note that the energy of the final [2 + 2]-cycloaddition product for MeC≡CMe is 15.2 kcal mol$^{-1}$ lower than the experimentally observed outcome of **3**, and so is thermodynamically favoured yet not observed. Furthermore, we tested reactions also with Bu$^t$CN, Bu$^t$NCO, and PhNCO and find no evidence of any reactivity. Lastly, we tested the reactivity of **2. tmeda** with one equivalent of PhN$_3$ in an attempt to prepare [U (Ts$^{Xy}$)(NPh)(tmeda)], but we find no evidence for the formation of this monomeric imido complex and in fact isolate only [{U (Ts$^{Xy}$)}$_2$(μ-η$^6$:η$^6$-C$_6$H$_5$Me)]. This suggests, in-line with calculations, that the monomeric imido version of **3** is thermodynamically high-lying and does not play a role in this chemistry. The combined lack of experimental evidence for the monomer-route reaction that is consistent with the computationally derived reaction profile thus rules out the monometallic reduction route for the direct four-electron reduction of PhNNPh, and suggests that **2**, **2.tmeda**, and **2.pmdeta** react by an oxidative addition route with PhNNPh.

## Discussion

Although interpretation of the reaction that affords **3** is convoluted by the fact that a dimeric formulation is observed, it is instructive to analyse the fundamental characteristics of this transformation. As unequivocally demonstrated by the combined structural, spectroscopic and magnetic characterisation data, each uranium centre has been formally oxidised by two units (i.e., (III) in **2** to (V) in **3**), the valence 5$f^n$ count is now 5$f^{n-2}$ (e.g., 5$f^3$ in **2** to 5$f^1$ in **3**), the metal valence count per uranium centre has increased by two (i.e., 9 in **2** to 11 in **3** discounting any π-bonding as is normal practice), and two new covalently bound ligand bonds have been installed in the coordination sphere of each uranium centre in **3**. Although **3** is dinuclear, from the perspective of each individual ion the transformation is clear-cut and since oxidative addition describes an overall transformation the reaction that produces **3** is thus a genuine, clear-cut oxidative addition, since it satisfies all the criteria for this reaction. This oxidative addition reaction is unique in actinide chemistry and contrasts to the previous multi-metal electron redox transformations described above[25–43]. There are few examples of low valent uranium complexes reacting with diazobenzene, and where documented the resulting di-imido complexes derived from a uranium(II) equivalent in a four-electron transformation[72],

cooperative uranium and non-innocent multi-electron redox couples involving charge loaded arenes[26,52,73,74], or no cleavage of the N=N bond occurs to give [LU(N$_2$Ph$_2$)] species where the diazobenzene retains a N–N bond and is formulated as a radical anion;[63] the latter is analogous to reactions of certain iridium complexes with dioxygen, where an O–O bond is retained and thus those reactions are not oxidative addition[3]. Further, alternative mechanisms that would invalidate a claim of oxidative addition are found to be unfeasible by experimentally supported computed reaction profiles.

Where reductive elimination is concerned, isotopic labelling studies suggest that this reaction is concerted since only isotopically pure H$_{10}$– and D$_{10}$–PhNNPh compounds are formed and no isotopic cross-over products are observed. Furthermore, an established nitrene trap produces no aziridine products when reactions are spiked, which suggests that nitrenes are not generated that itself is consistent with a concerted reductive elimination. Thus, even though the uranium by-product of the reductive elimination step remains inherently unknown, since the reaction mixture becomes an intractable mixture of products due to the thermal regime, all the experimental and computational evidence are internally consistent and uniformly point to a reductive elimination reaction since no other reaction could credibly account for the reformation of diazobenzene.

The evidence we have assembled for reversible oxidative addition and reductive elimination chemistry of **2**, **2.tmeda**, **2. pmdeta** and **3** advances the concept that these principal reaction types, which are key to classifying and understanding reactivity that has been prevalent and widely exploited in transition metal catalysis for over half a century, are feasible in f-block chemistry. This suggests that uranium can chemically mimic the d-block even though it is an actinide element. The question then arises as to why this system exhibits such reversible reactivity. This will certainly require further investigations, but some observations can be summarised at this juncture. The coordination of the Ts$^{Xy}$ ligand is quite open, which will allow substrates to enter and exit the coordination sphere of uranium straightforwardly. The ligand overall is quite rigid, so there would be anticipated to be minimal ligand-reorganisation energy that might be otherwise expected for a metal changing OS[75]. Despite the overall ligand rigidity, we note that because the N-aryl groups are planar and 'two-dimensional' the nitrogen centres can easily rotate from trigonal-planar to -pyramidal geometries, as found in **3**; they are thus in principle able to modulate their π-donor ability as required to meet the ligand donor requirements of the uranium ion as it shuttles from III to V OSs. Lastly, there are no other donor atoms in the Ts$^{Xy}$ ligand set other than the three amides to strongly favour metal high OSs compared to, for example, Tren ligands where the additional amine-anchor clearly stabilises high OS metal complexes and conversely destabilises low OS metal complexes.

The system reported here is clearly not optimised. However, the fact our combined experimental and computational evidence suggest that it can execute oxidative addition and be coerced into reductive elimination, with a substrate with a thermolytic disruption enthalpy of 93 kcal mol$^{-1}$[76], validates the notion that with suitable ancillary ligands uranium catalysis that exploits elementary oxidative addition and reductive elimination pathways centred on a uranium(III/V)-redox couple may well be achievable. With optimised supporting ligands that better-balance the redox couple the prospect that this could therefore form the basis of new catalytic cycles in f-block chemistry, for example the production of aniline derivatives, becomes realistic.

## Methods

**General**. Experiments were carried out under a dry, oxygen-free dinitrogen atmosphere using Schlenk-line and glove-box techniques. All solvents and reagents

were rigorously dried and deoxygenated before use. Compounds were variously characterised by elemental analyses, NMR, FTIR, EPR, and UV/Vis/NIR electronic absorption spectroscopies, Evans and SQUID magnetometric methods, single crystal X-ray diffraction studies, and DFT calculations. Further details are available in Supplementary Methods.

**Preparation of [U(Ts$^{Xy}$)(tmeda)] (2.tmeda).** A solution of TMEDA (0.46 g, 4.0 mmol) in hexanes (15 ml) was added to a cold (−78 °C) stirring mixture of **1** (1.78 g, 2.0 mmol) and KC$_8$ (0.30 g, 2.2 mmol). The stirring mixture was allowed to warm to room temperature slowly over 16 h. After this time, the purple solution was separated from the black precipitate by filtration through a fritted Schlenk, the solids washed with hexanes (3 × 5 ml), combined extracts reduced to dryness in vacuo to yield a purple solid. Recrystallisation of the solids from hot hexanes yielded pure **2.**tmeda as purple crystals. Purple block shaped crystals of **2.**tmeda suitable for X-ray diffraction studies were grown by storage of a saturated hexanes solution of **2.**tmeda at room temperature over 16 h. Yield 0.60 g, 33%. Anal. Calculated for C$_{37}$H$_{62}$N$_5$Si$_3$U: C, 49.42; H, 6.95; N, 7.79%. Found: C, 49.75; H, 7.01; N, 7.65%. $^1$H NMR (C$_6$D$_6$, 298 K): δ −43.60 (1H, s, Si-C$H$), −22.16 (1H, s, $p$-Ar-$H$), −19.49 (6H, s, $o$-Ar-$H$), −5.19 (12H, s, C$H_3$-TMEDA), −3.57 (18H, s, SiM$e_2$), -1.92 (1H, s, $p$-Ar-$H$), 1.96 (18H, s, C$H_3$), 3.80 (1H, s, $p$-Ar-$H$), 24.12 (4H, s, C$H_2$-TMEDA) ppm. FTIR ν cm$^{-1}$ (Nujol): 1600 (s), 1577 (vs), 1352 (m), 1326 (vs), 1308 (vs), 1242 (vs), 1182 (vs), 1167 (vs), 1029 (br, m), 1003 (br w), 974 (s), 890 (s), 847 (vs) 812 (vs), 774 (m), 740 (w), 704 (w), 671 (m), 640 (m), 582 (vw), 559 (vw), 529 (vw), 507 (vw). UV-vis λ$_{max}$/nm (ε/M$^{-1}$ cm$^{-1}$): 491 (793), 952 (137), 1045 (124), 1090 (117), 1125 (107), 1238 (103), 1303 (87), 1491 (86), 1552 (85). Magnetic moment (Evans method, C$_6$D$_6$, 298 K): μ$_{eff}$ = 2.70 μ$_B$.

**Preparation of [U(Ts$^{Xy}$)(pmdeta)] (2.pmdeta).** A solution of PMDETA (0.69 g, 4.0 mmol) in hexanes (15 ml) was added to a cold (−78 °C) stirring mixture of **1** (1.78 g, 2.0 mmol) and KC$_8$ (0.30 g, 2.2 mmol). The mixture was allowed to warm to room temperature slowly over 16 h. After this time, the purple solution was separated from the black precipitate by filtration through a fritted Schlenk, and the solids washed with hot hexanes (3 × 5 ml), and the volatiles were removed under reduced pressure to yield a dark purple pyrophoric solid. Recrystallization of this solid from hot hexanes yields pure **2.**pmdeta as dark violet crystals (1.07 g, 56%). Dark violet block shaped crystals of **2.**pmdeta suitable for X-ray diffraction studies were grown by storage of a saturated hexanes solution of **2.**pmdeta at room temperature over 16 h. Anal. Cald for C$_{40}$H$_{69}$N$_6$Si$_3$U: C, 50.24; H, 7.27; N, 8.79 %. Found: C, 50.05; H, 7.40; N, 8.42 %. $^1$H NMR (C$_6$D$_6$): δ 11.13 (3H, br s, C$H_3$ PMDTA), 2.20 (30H (12H + 18H), br s, C$H_3$ PMDTA, C$H_3$ ligand), −1.89 (8H, br s, C$H_2$ PMDTA), −2.69 (18H, br s, C$H_3$), −15.57 (6H, br s, $o$-C$H$), −17.77 (3H, br s, $p$-C$H$), −40.92 (1H, br s, Si-C$H$). FTIR ν cm$^{-1}$ (Nujol): 1599 (s), 1578 (vs), 1351 (m), 1305 (s), 1241 (s), 1177 (s), 1167 (vs), 1102 (w), 1032 (m), 1003 (w), 977 (m), 961 (m), 893 (s), 877 (vs), 858 (vs), 845 (vs), 814 (s), 774 (m), 741 (vw), 721 (vw), 708 (vw), 692 (vw), 669 (vw), 640 (m), 570 (vw). UV-vis (toluene): λ$_{max}$ (ε/M$^{-1}$ cm$^{-1}$): 947 (160), 1035 (115), 1084 (100), 1130 (80), 1230 (80), 1298 (60), 1496 (60). Magnetic moment (Evans method, C$_6$D$_6$, 298 K): μ$_{eff}$ = 2.69 μ$_B$.

**Preparation of [{U(Ts$^{Xy}$)(μ–NPh)}$_2$] (3).** Method A: Hexanes (2 ml) were added to a cold (−78 °C) stirring mixture of **2.**tmeda or **2.**pmdeta (0.9 mmol) and azobenzene (0.08 g, 0.5 mmol) in an ampoule. The resultant mixture was allowed to warm to room temperature over 16 h. After this time, hexanes (2 ml) were added and the mixture was heated and filtered while hot; the liquor was allowed to cool to room temperature and stored at room temperature for 16 h to yield crystals of **3**. The solid residue was recrystallised from hot toluene, filtered and allowed to cool to room temperature and stored at room temperature for 16 h also yields crystals of **3**. Both sets of crystals were isolated by filtration and dried by the passage of N$_2$ over the surface. Yield (crystalline combined): 0.37 g, 47 %. Further removal of solvent in vacuo was not achievable as **3** decomposes upon exposure to dynamic vacuum, but we note that **3** is thermally stable. Brown block shaped crystals of **3** suitable for X-ray diffraction studies were grown by storage of a saturated toluene solution of **3** at −30 °C over 16 h.

Method B: Hexanes (3 ml) were added to a cold (−78 °C) stirring mixture of [{U(Ts$^{Xy}$)}$_2$(μ-η$^6$:η$^6$-C$_6$H$_5$Me)] (0.83 g, 0.5 mmol) and azobenzene (0.09 g, 0.5 mmol) in an ampoule. The resultant mixture was allowed to warm to room temperature over 16 h. After this time, the hexanes (2 ml) were added and the mixture was heated and filtered whilst hot. Toluene (2 ml) was added to the residual solids and was heated and filtered whilst hot. Both solutions were stored at room temperature for 16 h and crystals of **3** were deposited in both. The crystals were isolated by filtration and dried by the passage of N$_2$ over them. Yield (crystalline combined): 0.25 g, 29%. Further removal of solvent in vacuo was not achievable as **3** decomposes upon exposure to vacuum. Brown block shaped crystals of **3** suitable for X-ray diffraction studies were grown by storage of a saturated toluene solution of **3** at room temperature over 16 h.

Method C: Hexanes (3 ml) were added to a cold (−78 °C) stirring mixture of **1** (0.89 g, 1.0 mmol), KC$_8$ (0.14 g, 1.0 mmol) and azobenzene (0.09 g, 0.5 mmol) in an ampoule. The resultant mixture was allowed to warm to room temperature over 16 h. After this time, the mixture was heated and filtered while hot and the liquor was allowed to cool to room temperature and stored at room temperature for 16 h to yield crystals of **3**. The solid residue was recrystallised from hot toluene (3 ml), filtered and allowed to cool to room temperature and stored at room temperature for 16 h also yielding crystals of **3**. Both sets of crystals were isolated by filtration and dried by the passage of N$_2$ over the surface. Yield (crystalline combined): 0.37 g, 47 %. Anal. Calculated for C$_{74}$H$_{102}$N$_8$Si$_6$U$_2$·1.05C$_7$H$_8$: C, 52.96; H, 6.03; N, 6.07 %. Found: C, 53.30; H, 5.97; N, 6.46 %. $^1$H NMR (C$_6$D$_6$): δ 13.87 (4H, br s, $o$-Ar-H NPh), 6.67 (2H, t $^3J_{HH}$ = 8.0 Hz, $p$-Ar-H NPh), 4.72 (4H, t $^3J_{HH}$ = 6.9 Hz, $m$-Ar-H NPh), 4.58 (6H, br s, $p$-Ar-H Ts$^{Xy}$), 2.52 (12H, br s, $o$-Ar-$H$ Ts$^{Xy}$), 1.50 (36H, br s, C$H_3$), −0.56 (36H, br s, SiM$e_2$), −23.45 (2H, br s, SiC$H$) ppm. FTIR ν/cm$^{-1}$ (Nujol): 1601 (vs), 1353 (s), 1328 (s), 1292 (m), 1253 (s), 1179 (s), 1153 (m), 1031 (m), 999 (s), 956 (m), 895 (m), 850 (vs), 825 (vs), 774 (m), 753 (w), 722 (m), 649 (m), 581 (vw), 561 (vw). UV-vis λ$_{max}$/nm (ε/M$^{-1}$ cm$^{-1}$): 1011 (185), 1230 (147), 1527 (130). Magnetic moment (Evans method, THF-d$_8$, 298 K): μ$_{eff}$ = 3.12 μ$_B$.

**Preparation of [{U(Ts$^{Xy}$)(μ–NPh-d$_5$)}$_2$] (3-D$_{10}$).** Pentane (15 ml) was added to a cold (−78 °C) mixture of [{U(Ts$^{Xy}$)}$_2$(μ-η$^6$:η$^6$-C$_6$H$_5$Me)] (1.09 g, 0.7 mmol) and D$_{10}$-azobenzene (0.13 g, 0.7 mmol). The resultant mixture was allowed to warm to room temperature over 16 h. After this time volatiles were removed in vacuo and the dark solid obtained extracted with 10 ml hot (80 °C) toluene, filtered while hot and the liquor was allowed to cool to room temperature and stored at −30 °C for 16 h to yield crystals of 3-D$_{10}$. The solid residue was recrystallised from hot toluene, filtered and allowed to cool to room temperature and stored at room temperature for 16 h also yielding crystals of 3-D$_{10}$. Both sets of crystals were isolated by filtration and dried in vacuo. Yield (crystalline combined): 0.35 g, 30%. Brown block shaped crystals of 3-D$_{10}$ suitable for X-ray diffraction studies were grown by storage of a saturated toluene solution of 3-D$_{10}$ at −30 °C over 16 h. Anal. Calculated for C$_{74}$H$_{92}$D$_{10}$N$_8$Si$_6$U$_2$: C, 50.55; H, 5.27; N, 6.37 %. Found: C, 51.02; H, 5.29; N, 6.28 %. $^1$H NMR (C$_6$D$_6$): δ 4.59 (6H, br s, $p$-Ar-H Ts$^{Xy}$), 2.45 (12H, br s, $o$-Ar-H Ts$^{Xy}$), 1.50 (36H, br s, C$H_3$), −0.61 (36H, br s, SiM$e_2$), −23.90 (2H, br s, SiC$H$) ppm. FTIR ν/cm$^{-1}$ (Nujol): 1595 (m), 1580 (s), 1459 (m), 1349 (m), 1288 (s), 1247 (s), 1161 (s), 1147 (s), 1029 (m), 975 (s), 949 (m), 887 (m), 872 (m), 8471 (vs), 829 (vs), 809 (vs), 774 (s), 751 (s), 697 (s), 671 (s), 647 (s), 629 (s), 593 (w), 584 (w), 567 (m), 549 (s), 487 (s), 471 (s), 437 (m). UV-vis λ$_{max}$/nm (ε/M$^{-1}$ cm$^{-1}$): 1021 (188), 1234 (170), 1538 (160). Magnetic moment (Evans method, C$_6$D$_6$, 298 K): μ$_{eff}$ = 3.78 μ$_B$.

**Extrusion of azobenzene by reductive elimination of 3.** Complex **3** was placed in the end bulb of a two bulb sublimation tube and the other bulb was cooled with liquid nitrogen. The reductive elimination product, azobenzene, was collected in the cooled bulb as an orange solid by heating the sample at 100 °C, 4 × 10$^{-6}$ mbar. Yield: 0.07 g, 26%. Analysis of azobenzene: $^1$H NMR (C$_6$D$_6$): δ 8.01 (4H, m, CH), 7.17–7.09 (6H, m, CH). GC-MS (ESI positive, MeOH): m/z 182.1 (32%) {PhNNPh}$^+$; (ESI negative, MeOH): m/z 180.97 (64%) {PhNNPh−H$^+$}$^-$.

**Extrusion of azobenzene/D$_{10}$-azobenzene by reductive elimination of a 50:50 mixture of 3/3-D$_{10}$.** An equimolar mixture of **3** and 3-D$_{10}$ was placed in the end of a sublimation tube. The reductive elimination products, azobenzene/D$_{10}$-azobenzene, were collected in a cooled section further along the tube as an orange solid by heating the sample at 180 °C/10$^{-6}$ mbar. Yield: 0.06 g, 23%. Analysis of azobenzene/D$_{10}$-azobenzene: $^1$H NMR (CD$_3$CN): δ 7.93–7.90 (4H, m, CH), 7.61–7.55 (6H, m, CH) ppm. $^{13}$C{$^1$H} NMR (CD$_3$CN): δ 153.11 (s, i-C-D$_{10}$-azobenzene), 153.02 (s, i-C-azobenzene), 131.88 (s, p-CH), 131.38 (t $J_{CD}$ = 25.4 Hz, p-CD), 129.93 (s, o-CH), 129.43 (t $J_{CD}$ = 24.4 Hz, o-CD), 123.14, (s, m-CH), 122.74 (t $J_{CD}$ = 25.4 Hz, m-CD) ppm. Mass spectrometry (ESI positive, MeOH): m/z 183.0 (62%) {PhNNPhH$^+$}$^+$, 193.1 (75%) {D$_{10}$-PhNNPhH}$^+$; (ESI negative, MeOH): m/z 181.8 (100%) {PhNNPh−H$^+$}$^-$, 193.0 (60%) {D$_{10}$-PhNNPhH}$^-$, 223.8 (45%) {D$_{10}$−PhNNPh·MeOH−H}$^-$.

**Attempted reactions of 2.tmeda, 2.pmdeta, or 3 with alkynes, nitriles, and isocyanates – representative procedure.** On a 0.5 mmol scale with respect to uranium, a 1:2 solution of azobenzene:substrate (substrate = alkyne, nitrile, or isocyanate) in toluene (5 ml) was added to a cold (-78 °C) solution of **2.**tmeda or **2.** pmdeta, or two equivalents of substrate were added to **3**, each in 10 ml toluene. The resulting mixture was allowed to warm to room temperature while stirring and stirred for a further 16 h. No reaction was observed as monitored by $^1$H NMR spectroscopy of an aliquot of the reaction mixture. The same result was obtained after the solution was heated up to 80 °C for 12 h. Reagents attempted: MeC≡CMe, PhC≡CPh, Bu$^t$C≡CH, Me$_3$SiC≡CH, Bu$^t$C≡P·HMDSO, Bu$^t$CN, Bu$^t$NCO, and PhNCO.

**Attempted preparation of [U(Ts$^{Xy}$)(NPh)(tmeda)].** Compound **2.**tmeda (0.45 g, 0.5 mmol) was treated with one equivalent of PhN$_3$ (0.5 M solution in Bu$^t$OMe) in toluene (5 ml). Overnight storage produced crystals that were determined to be [{U(Ts$^{Xy}$)}$_2$(μ-η$^6$:η$^6$-C$_6$H$_5$Me)]. No other products could be identified from this reaction. Repeating the reaction in pentane gave an insoluble mixture that when dissolved in D$_8$-toluene gave an identical result.

**Data availability.** The X-ray crystallographic coordinates for structures reported in this Article have been deposited at the Cambridge Crystallographic Data Centre

(CCDC), under deposition number CCDC 1529405–1529408. These data can be obtained free of charge from The Cambridge Crystallographic Data Centre via www.ccdc.cam.ac.uk/data_request/cif, respectively. All the other data are available from the corresponding authors upon request.

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

## Acknowledgements

We are grateful to the Royal Society (grant UF110005), European Research Council (grant CoG612724), UK Engineering and Physical Sciences Research Council (grants EP/G051763/1 and EP/M027015/1), the Humboldt Foundation, the National UK EPR Facility at Manchester, Universities of Manchester and Nottingham, UK National Nuclear Laboratory, COST Action CM1006, and CalMip for generous funding and support.

## Author contributions

B.M.G., D.P., and E.L.: Conducted the synthetic work and analysed the data. C.E.K. and L.M.: Conducted and analysed the reaction profile calculations. F.T. and E.J.L.M.: Conducted and analysed the EPR and magnetic measurements. A.J.W.: Conducted crystallographic refinements. S.T.L.: Originated the central idea, supervised the work, analysed the data, and wrote the manuscript with input from all the authors.

## Additional information

**Competing interests:** The authors declare no competing financial interests.

