## [Peer Review File · Nature Communications]

Reviewers' comments:

Reviewer #1 (Remarks to the Author):

In this manuscript, Gardner et al. describe their discovery that two-electron oxidative addition and reductive elimination can occur at uranium. This is a remarkable observation as such processes are generally only observed by d-block transition metals.

Comments:

-The manuscript is very well-written, is clear and concise, and is well organized.
-I believe that the authors conclusions are adequately supported by the data.
-My only real reservation about this article regards its overall impact. The observation is undoubtedly important to the community; however, no insight is provided as to why this complex does this two-electron chemistry while others do not. Is it all about the ligand? If so, what aspects of the ligand design impart this reactivity. Furthermore, the reaction presented (using azobenzene) is not very interesting or broadly applicable. Yet, let me restate that this is remarkable chemistry.

Revisions:

-the authors should include ^{13}C NMR spectra for their complexes
-Why are the calculations not reported in free energy (dG) rather than enthalpy (dH). Because of the nature of this unique reaction, entropy certainly plays a role.
-regarding the above last comment, some insight as to why this complex displays two-electron OA/RE chemistry should be included rather than just supporting that it did happen.

Reviewer #2 (Remarks to the Author):

The manuscript describes reaction of previously reported sterically unsaturated and electron-rich uranium(III) compounds with azobenzene, yielding a dimeric uranium(V) imido species. This reaction is not surprising by itself; indeed, a number of examples now exist of the cleavage of azobenzene and related species by uranium, as indicated by the list of references cited.

What is novel in this report, however, is the facile reversibility of this oxidation to yield U(III) through extrusion of azobenzene under mild conditions (vacuum, mild heating). Although other reports have demonstrated pathways to the reduction of such imido species, to my knowledge this is the first example of spontaneous reduction of this nature.

The authors acknowledge that the net reactions are two-electron oxidation addition and reductive elimination, while specifying that this description has "no mechanistic implications". Theoretical treatment of the reaction pathway for the bare metal center suggests reaction of an initially generated uranium (IV) adduct of singly-reduced azobenzene (perhaps in equilibrium with a U(V) hydrazinato species) with a second equivalent of the U(III) synthon gives rise to the uranium(V) imido dimer.

Support for a concerted mechanism in the reduction and extrusion of azobenzene is provided by the lack of crossover in an experiment in which the imido compound and a derivative containing D10 azobenzene undergo reduction. This also is indirect support for a lack of a monomer-dimer equilibrium in solution, as presumably some scrambling to yield 3-D5 would occur, and no mixed D5-PhNNPh is observed, although use of a mixed PhNNAr azobenzene reagent would also help substantiate these mechanisms; I would encourage this be examined.

My only quibble with the mechanistic discussion is the argument that lack of observed trapping of a imidouranium species with alkynes is good evidence that the oxidation addition does not occur via initial formation of a monomeric uranium(V) species. It is far more common to trap imidouranium species by the use of isocyanates or nitriles, where heteroatom coordination

predisposes adduct formation with the trapping agent. I would suggest that these reactions be reported.

Overall, the manuscript describes a novel reaction of uranium; it remains to be seen, of course, if it can be included in a productive catalytic reaction. The work is well executed and the compounds well characterized. The result is worthy of publication, although I would prefer to see the results of the above-suggested additional experiments in trapping.

Reviewer #3 (Remarks to the Author):

Manuscript NCOMMS-17-11819.

The goal of this article is to demonstrate for the first time that a two electron oxidative addition followed by a two electron reductive elimination on a single metal ion is an accessible process for an f-element. While it is a common redox process within the d-transition metals and at the basement of most catalytic cycles, it has never been observed with the f-elements. With the choice of uranium for which a palette of oxidation states is available (from +II to +VI), the team of Liddle claims that the oxidation of trivalent (TsXy)U(L) complexes with azobenzene which affords a bimetallic imido U(V) species is a reversible pathway due to reductive elimination of azobenzene. If true, such reaction would give another dimension for the f-elements for which such process was believed unreachable and would offer new opportunities in catalysis.

This paper is a nice piece of work. The manuscript is well written. The experimental data are clear and of confidence since the new compounds are thoroughly characterized by an impressive battery of techniques (NMR, X-ray crystal structure, molecular magnetic measurements, UV-vis spectra, electropray). Hypotheses on the mechanism are underpinned with DFT calculations.

Concerning the oxidation of U(III) complexes with azobenzene, the reaction is classical and has been studied in a number of groups (C. Burns, W. J. Evans, Cummins, S. Bart...) with either 1e reduction of the ligand (radical PhN-NPh- : Coronado(2015); with Ln : Evans(2016), K.Mashima(1997),....), 2e reduction (PhN-NPh2-, Evans 1986 and 1988 with Sm(II)) or 4e reductive cleavage of the N=N double bond to the formation of imido compounds (Burns 1998, Evans 2008, Bart 2016).

In this manuscript, oxidation of the two U(III) adducts into the stable pentavalent bimetallic species 3 bridged by two PhN2-imido groups is clear and not confusing. Similar bis-imido uranium(IV or V) species have already been obtained in the groups of R. A. Andersen (1988), Burns(1999), Berthet (2008), Diaconescu (2010), Cummins (2010, 2012), Boncella (2014), Liddle (2014) and Mazzanti (2016). All the imido species were stable and evolution with releasing of azobenzene was never mentioned.

If the 2e -oxidation of a X3U into X3U(Y)2 is unique, complex 3 (as the authors say) would not result from a transient imido [(TsXy)U(=NPh)] species as this argument is refuted by DFT calculation and chemical attempts at trapping it. However, instead or in addition to hypotheses and attempts at trapping the terminal imido species [(TsXy)U(=NPh)], perhaps formation of genuine [(TsXy)U(L)(=NPh)] (L = tmeda, pmdeta or nothing) by treatment of [(TsXy)U(L)] (L = tmeda, possible pmdeta) with PhN3 should be attempted. If stable, its reactivity could be studied and offer missing information; although sterically insaturated, can [(TsXy)U(L)] (see Inorg Chim Acta 263 (1997) 171-180) or its more congested analogues [(TsXy)U(L)(=NPh)] be isolated and do they move directly into pentavalent 3? If isolable does [(TsXy)U(L)(=NPh)] also release azobenzene by warming under vacuum?

The proof for a 2e electron reductive elimination is the point that is the less convincing. By heating under vacuum, the orange azobenzene is collected and perfectly characterized. But, in contrast to what schematized in Figure 1c, there is no evidence that the release of PhN=NPh from 3 leads to

trivalent compound 2 which is described unstable affording the dinuclear imido aryl bridged species as in ref 47. May be the azobenzene releasing follows another course of reaction with degradation of the uranium complex without passage from the U(III) species 2 and then the theory of the authors is undermined. The authors don't give any details on the uranium product resulting from the extrusion of azobenzene under vacuum; formation of complex 2 (or its evolution product) is not probed so that it is difficult to ensure that the 2e reductive elimination process occurs as suggested. Computed profile (Figure 5) corroborates the difficulty of the reverse elimination reaction. The enthalpy of formation of the U(V) complex 3 is -42kcal/mol, far below that of the trivalent complex 2 and the corresponding ΔG (passage from 2 \rightarrow 3) also close to -47 kcal/mol meaning that the reverse reaction is thermodynamically improbable.

It is why I can not accept this article to nature comm.

Questions :

Can the authors bring information on the uranium product formed during heating of 3 and extrusion of azobenzene ?

As complex 3 readily decomposes upon exposure to vacuum, what are the behaviour and stability of 3 in polar or non-polar solvents when heated ?

Is there an equilibrium in solution between the imido 3 and U(III) species + azobenzene ?

I think that the two following articles should be in the references:

- JACS 199, 121, 5585 : although there is no reductive elimination as in the present manuscript, C. Burns et al reported a 2-electron (+4/+6) redox processes in the catalytic reduction of azides and hydrazines with U(VI) organouranium complexes.

- Eur. J. Inorg. Chem. 2004, 1996 ; Mehdoui et al reported a reversible 1e (per U) oxidative addition-reductive elimination of pyrazine on U(III) complexes

Some points require changes:

Page 4-5 : when the authors said that "(i) cooperative multi-metallic redox transformations [Type (a) in Fig. 1] utilising multiple single electron uranium redox couples [U(III) to U(IV) or U(V) to U(VI)] where one new covalent uranium-ligand bond per uranium centre is formed", there are also examples of U(III)- \rightarrow U(V) (oxo and imido species form with amine oxide and organic azides) and U(IV)- \rightarrow U(VI) (see JACS (1992),114, 10068) oxidation reactions.

Page 8 : "The electronic absorption spectrum of 3 in toluene exhibits broad absorptions at 6420, 7180, and 8460 cm^{-1} ($\epsilon = 40\text{-}70 \text{ M}^{-1} \text{ cm}^{-1}$) in the NIR region"

\ \rightarrow These bands are not visible from the spectrum in SI

Page 11 : "with respect to free-azobenzene (1.338 Å)"

The N=N distance seems quite long:

\ \rightarrow 1.244Å from J. Fluorine Chem. (2005), 126, 515

\ \rightarrow 1.249 from JACS(2004),126, 3539

Page 13 : "There are few examples of low valent uranium complexes reacting with diazobenzene, and where documented the resulting di-imido complexes derived from cooperative uranium and non-innocent multielectron redox couples involving charge loaded arenes..."

\ \rightarrow The sentence is incorrect; indeed, in the reaction of $[\text{Cp}^*2\text{UCl}_2][\text{Cl}]$ with $\text{PhN}=\text{NPh}$ giving the U(VI) diimido complex $[\text{Cp}^*2\text{U}(=\text{NPh})_2]$ there is no charge loaded arene ligand coordinated to the metal. See Angew. Chem. Int. Ed. 1998, 37, No. 7, 959.

Reviewer #4 (Remarks to the Author):

The report only considers the crystallographic aspects of the submission.

Four structures are reported, two of which are of isostructural. The structures are well determined

and details have been provided in the CIFs concerning the modelling of disorder and solvent treatment through SQUEEZE. No changes to the structures reported are required.

Although documented that this is the cause of the discrepancy in the calculated and reported formula, the standard practice has been changed so the solvent accounted for by Squeeze is no longer given explicitly in the formula.

Otherwise the only issue leading to A and B alerts in the checkcif report is that there are significant residual electron density peaks and troughs around the heavy atom. While it is not unreasonable that there are large peaks adjacent to heavier atoms that are only a small percentage of the total electron density of the U atom, the holes in the difference Fourier maps are larger than expected for 2.pmdeta. It is not suggested that structure is incorrect or action required in this case but in future it may be that it could be improved by re-examination of the diffraction data to see if the crystal was split or twinning is present that has not been resolved. In general F_{obs} is greater than F_{calc} for this refinement and there are many high angle systematic absences with significant intensity which would be consistent with contributions from another component or the data collection or integration parameters being set such that intensity from adjacent reflections was included.

I suggest modifying the text describing the structures of 2.tmeda and 2.pmdeta given on page 7 of the manuscript. The authors highlight how the structure appears to have a very exposed U ion without the stabilising amine but this could be said of many structures if you virtually remove a significant part of the co-ordination sphere of an ion. As shown by the structures of 3 the formation of dimers is also a way to complete coordination. The authors should clarify exactly what they mean to highlight here and change the text accordingly.

We thank the reviewers for their considered comments. We have endeavoured to address all the additional suggested experiments, within the inherent limitations of the system as we outlined in our initial submission, and address the reviewer comments on a point-by-point basis below.

Reviewers' comments:

Reviewer #1 (Remarks to the Author):

In this manuscript, Gardner et al. describe their discovery that two-electron oxidative addition and reductive elimination can occur at uranium. This is a remarkable observation as such processes are generally only observed by d-block transition metals.

Comments:

-The manuscript is very well-written, is clear and concise, and is well organized.
-I believe that the authors conclusions are adequately supported by the data.
-My only real reservation about this article regards its overall impact. The observation is undoubtedly important to the community; however, no insight is provided as to why this complex does this two-electron chemistry while others do not. Is it all about the ligand? If so, what aspects of the ligand design impart this reactivity. Furthermore, the reaction presented (using azobenzene) is not very interesting or broadly applicable. Yet, let me restate that this is remarkable chemistry.

Revisions:

-the authors should include ^{13}C NMR spectra for their complexes.

RESPONSE: Sadly we have not been able to acquire ^{13}C NMR spectra of the complexes in this paper. The uranium(III) complexes are too sensitive and decompose before ^{13}C NMR spectra can be collected due to the long accumulation times. Neither of the imido dimers give any peaks in their ^{13}C NMR spectra, even with long relaxation times and 1024 scans, which takes a long time to accumulate, due to the fact these complexes are only sparingly soluble once isolated in crystalline form and they precipitate out during measurements.

-Why are the calculations not reported in free energy (dG) rather than enthalpy (dH). Because of the nature of this unique reaction, entropy certainly plays a role.

RESPONSE: Indeed, the entropy may play a role in the reaction, but the entropy contribution is so far not correctly computed so that the error introduced on the ΔG value is very important. Therefore, several authors have proposed to discuss only the computed enthalpy as there is much less error and as it normally does not change the chemistry. Moreover, a recent contribution from Castr *et al.* (ACS Catalysis **2015**, 5(1), 416-425) have shown that the ΔH was leading to equivalent results as the ΔG in solution including dispersion. Therefore, we prefer to stick to the ΔH profiles.

-regarding the above last comment, some insight as to why this complex displays two-electron OA/RE chemistry should be included rather than just supporting that it did happen.

RESPONSE: The referee raises a good and interesting point. We have thus added the following text to the discussion section which avoids speculation and summarises some known facts: "*The question then arises as to why this system exhibits such reversible reactivity. This will certainly require further investigations, but some observations can be summarised at this juncture: (i) the coordination of the Ts^{xy} ligand is quite open allowing substrates to enter and exit the coordination sphere of uranium straightforwardly; (ii) the ligand overall is quite rigid so there would be anticipated to be minimal ligand-reorganisation energy;⁷⁵ (iii) despite the overall ligand rigidity we note that because the N-aryl groups are planar and 'two-dimensional' the nitrogen centres can easily rotate from trigonal-planar to -pyramidal geometries, thus modulating their π -donor ability as required to meet the oxidation state requirements of the uranium ion as it shuttles from III to V states; (iv) there are no other donor atoms in the Ts^{xy} ligand set other than the three amides to favour metal high oxidation states compared to, for example, Tren ligands where the additional amine clearly stabilises high oxidation state metal complexes.*"

Reviewer #2 (Remarks to the Author):

The manuscript describes reaction of previously reported sterically unsaturated and electron-rich uranium(III) compounds with azobenzene, yielding a dimeric uranium(V) imido species. This reaction is not surprising by itself; indeed, a number of examples now exist of the cleavage of azobenzene and related species by uranium, as indicated by the list of references cited.

What is novel in this report, however, is the facile reversibility of this oxidation to yield U(III) through extrusion of azobenzene under mild conditions (vacuum, mild heating). Although other reports have demonstrated pathways to the reduction of such imido species, to my knowledge this is the first example of spontaneous reduction of this nature.

The authors acknowledge that the net reactions are two-electron oxidation addition and reductive elimination, while specifying that this description has "no mechanistic implications". Theoretical treatment of the reaction pathway for the bare metal center suggests reaction of an initially generated uranium (IV) adduct of singly-reduced azobenzene (perhaps in equilibrium with a U(V) hydrazinato species) with a second equivalent of the U(III) synthon gives rise to the uranium(V) imido dimer.

Support for a concerted mechanism in the reduction and extrusion of azobenzene is provided by the lack of crossover in an experiment in which the imido compound and a derivative containing D10 azobenzene undergo reduction. This also is indirect support for a lack of a monomer-dimer equilibrium in solution, as presumably some scrambling to yield 3-D5 would occur, and no mixed D5-PhNNPh is observed, although use of a mixed PhNNAr azobenzene reagent would also help substantiate these mechanisms; I would encourage this be examined.

RESPONSE: By NMR we observe no evidence for a monomer-dimer equilibrium. We also feel the asymmetric azobenzene would not actually add anything significant since it would effectively be reproducing the conclusion we get from using labeled azobenzene. Further, as we mention below nitriles and isocyanates do not trap out a monomeric imido or disrupt the imido dimers, even on extended heating at 80 °C for 12 hours, which is in-line with our computational work, so we feel this extra experiment, which would take a long time to conduct, would not actually add anything significant to the paper.

My only quibble with the mechanistic discussion is the argument that lack of observed trapping of a imidouranium species with alkynes is good evidence that the oxidation addition does not occur via initial formation of a monomeric uranium(V) species. It is far more common to trap imidouranium species by the use of isocyanates or nitriles, where heteroatom coordination predisposes adduct formation with the trapping agent. I would suggest that these reactions be reported.

RESPONSE: We have attempted the reactions with Bu^tCN, Bu^tNCO, and PhNCO. No involvement of those reagents was observed. This certainly argues against a monomeric imido being present, which is also in-line with the observed reactivity with PhN₃ (see below) and we have added some text addressing this point at the end of the mechanistic study.

Overall, the manuscript describes a novel reaction of uranium; it remains to be seen, of course, if it can be included in a productive catalytic reaction. The work is well executed and the compounds well characterized. The result is worthy of publication, although I would prefer to see the results of the above-suggested additional experiments in trapping.

Reviewer #3 (Remarks to the Author):

The goal of this article is to demonstrate for the first time that a two electron oxidative addition followed by a two electron reductive elimination on a single metal ion is an accessible process for an f-element. While it is a common redox process within the d-transition metals and at the basement of most catalytic cycles, it has never been observed with the f-elements. With the choice of uranium for which a palette of oxidation states is available (from +II to +VI), the team of Liddle claims that the oxidation of trivalent (TsXy)U(L) complexes with azobenzene which affords a bimetallic imido U(V) species is a reversible pathway due to reductive elimination of azobenzene. If true, such reaction would give another dimension for the f-elements for which such process was believed unreachable and would offer new opportunities in catalysis.

This paper is a nice piece of work. The manuscript is well written. The experimental data are clear and of confidence since the new compounds are thoroughly characterized by an impressive battery of techniques (NMR, X-ray crystal structure, molecular magnetic measurements, UV-vis spectra, electropray). Hypotheses on the mechanism are underpinned with DFT calculations.

Concerning the oxidation of U(III) complexes with azobenzene, the reaction is classical and has been studied in a number of groups (C. Burns, W. J. Evans, Cummins, S. Bart...) with either 1e reduction of the ligand (radical PhN-NPh⁻ : Coronado(2015); with Ln : Evans(2016), K.Mashima(1997),...), 2e reduction (PhN-NPh²⁻, Evans 1986 and 1988 with Sm(II)) or 4e reductive cleavage of the N=N double bond to the formation of imido compounds (Burns 1998, Evans 2008, Bart 2016).

In this manuscript, oxidation of the two U(III) adducts into the stable pentavalent bimetallic species 3 bridged by two PhN₂-imido groups is clear and not confusing. Similar bis-imido uranium(IV or V) species have already been obtained in the groups of R. A. Andersen (1988), Burns(1999), Berthet (2008), Diaconescu (2010), Cummins (2010, 2012), Boncella (2014), Liddle (2014) and Mazzanti (2016). All the imido species were stable and evolution with releasing of azobenzene was never mentioned.

If the 2e -oxidation of a X₃U into X₃U(Y)₂ is unique, complex 3 (as the authors say) would not result from a transient imido [(TsXy)U(=NPh)] species as this argument is refuted by DFT calculation and chemical attempts at trapping it. However, instead or in addition to hypotheses and attempts at trapping the terminal imido species [(TsXy)U(=NPh)], perhaps formation of genuine [(TsXy)U(L)(=NPh)] (L = tmeda, pmdeta or nothing) by treatment of [(TsXy)U(L)] (L = tmeda, possible pmdeta) with PhN₃ should be attempted. If stable, its reactivity could be studied and offer missing information; although sterically unsaturated, can [(TsXy)U(L)] (see Inorg Chim Acta 263 (1997) 171-180) or its more congested analogues [(TsXy)U(L)(=NPh)] be isolated and do they move directly into pentavalent 3? If isolable does [(TsXy)U(L)(=NPh)] also release azobenzene by warming under vacuum?

RESPONSE: We performed the reaction of [U(Ts^{Xy})(tmeda)] (2.tmeda) with one equivalent of PhN₃ in toluene or pentane then toluene (for solubility reasons to get NMR data) and to our surprise isolated crystals of [(U(Ts^{Xy}))₂(μ-η⁶:η⁶-C₆H₅Me)] on both occasions. This suggests that the arene is thermodynamically downhill compared to the putative imido monomer, but we know that the arene reacts to give the imido dimer. This is further evidence that the imido monomer is a high-lying, experimentally unfeasible product in this scenario. We have added a description of this to the manuscript. In general, the monomeric formulation has eluded all attempts to make it, and the imido dimer formulation from diazobenzene seems to be the product of reactions always. Even chelating ligands such as tmeda and pmdeta cannot enforce a monomeric formulation, and this is in-line with the computed reaction mechanism. This is probably because the 'open' nature of the TsXy ligand enables the U₂N₂ ring to form without any steric congestion and multiple anionic σ-bonds will out-compete σ/π-combinations and dative donors.

The proof for a 2e electron reductive elimination is the point that is the less convincing. By heating under vacuum, the orange azobenzene is collected and perfectly characterized. But, in contrast to what schematized in Figure 1c, there is no evidence that the release of PhN=NPh from 3 leads to trivalent compound 2 which is described as unstable affording the dinuclear imido aryl bridged species as in ref 47. Maybe the azobenzene releasing follows another course of reaction with degradation of the uranium complex without passage from the U(III) species 2 and then the theory of the authors is undermined. The authors don't give any details on the uranium product resulting from the extrusion of azobenzene under vacuum; formation of complex 2 (or its evolution product) is not probed so that it is difficult to ensure that the 2e reductive elimination process occurs as suggested. Computed profile (Figure 5) corroborates the difficulty of the reverse elimination reaction. The enthalpy of formation of the U(V) complex 3 is -42kcal/mol, far below that of the trivalent complex 2 and the corresponding DG (passage from 2 -> 3) also close to -47 kcal/mol meaning that the reverse reaction is thermodynamically improbable.

It is thus quite difficult to accept this article to nature comm

RESPONSE: As we believed we had made clear we have made strenuous attempts to identify the resulting uranium product from the elimination step, but have never been successful in that regard because it is an inherent limitation of the system, which is why we have conducted labeling and trapping experiments. The calculated data are in our view entirely consistent with the fact that the mixture needs to be heated to extrude azobenzene.

Questions :

Can the authors bring information on the uranium product formed during heating of 3 and extrusion of azobenzene ?

RESPONSE: We addressed this point above. It is a natural limitation of the system so we feel we cannot do any more in that regard, which is why we only claim evidence and do not make an absolute claim.

As complex 3 readily decomposes upon exposure to vacuum, what are the behaviour and stability of 3 in polar or non-polar solvents when heated ?

RESPONSE: There appears to be little change of this species in solution when heated, polar or non-polar. The dimeric formulation seems to be quite robust and solvent molecules do not seem to compete for coordination to uranium against the imido groups.

Is there an equilibrium in solution between the imido 3 and U(III) species + azobenzene ?

RESPONSE: No, we have not found any evidence for an equilibrium in solution, which is consistent with the need to heat to push the reaction backwards.

I think that the two following articles should be in the references:

- JACS 199, 121, 5585 : although there is no reductive elimination as in the present manuscript, C. Burns et al reported a 2-electron (+4/+6) redox processes in the catalytic reduction of azides and hydrazines with U(VI) organouranium complexes.

- Eur. J. Inorg. Chem. 2004, 1996 ; Mehdoui et al reported a reversible 1e (per U) oxidative addition-reductive elimination of pyrazine on U(III) complexes .

RESPONSE: We have now included these two citations. They do not impinge on any priority claims in this paper.

Some points require changes:

Page 4-5 : when the authors said that "(i) cooperative multi-metallic redox transformations [Type (a) in Fig. 1] utilising multiple single electron uranium redox couples [U(III) to U(IV) or U(V) to U(VI)] where one new covalent uranium-ligand bond per uranium centre is formed", there are also examples of U(III)-> U(V) (oxo and imido species form with amine oxide and organic azides) and U(IV)-> U(VI) (see JACS (1992),114, 10068) oxidation reactions.

RESPONSE: This is an interesting point, which we are certainly sympathetic to. However, the quoted text comes from a passage relating to oxidative addition-type behavior, as we stated, whereas the reactions mentioned are certainly two-electron oxidations only one new ligand enters the coordination sphere of uranium so they are not oxidative addition(-type). That said we agree this point should be more explicitly acknowledged so we have inserted the following text in that section "*which are distinct to two-electron oxidations of uranium to give terminal mono-oxo and -imido ligands*". This then brings the questions of which references to cite to support that statement. There are quite a few examples in the literature, and we have a reference limit, which we are already exceeding, so to cover this we cite two reviews that cover all such examples.

Page 8 : "The electronic absorption spectrum of 3 in toluene exhibits broad absorptions at 6420, 7180, and 8460 cm⁻¹ (ϵ = 40-70 M⁻¹ cm⁻¹) in the NIR region"

è ->These bands are not visible from the spectrum in SI

RESPONSE: We thank the reviewer for spotting this transcribing error; the correct values are 6570, 7650, and 9815 cm⁻¹ and this correction has been made.

Page 11 : "with respect to free-azobenzene (1.338 Å)"

The N=N distance seems quite long:

è ->1.244Å from J.Fluorine Chem. (2005), 126, 515

è ->1.249 from JACS(2004),126, 3539

RESPONSE: The distance referred to is that of computed PhNNPh. However the reviewer makes a fair point, so we

have clarified that and also provided the range of N=N distances in crystal structures of free azobenzene in the CSD.

Page 13 : "There are few examples of low valent uranium complexes reacting with diazobenzene, and where documented the resulting di-imido complexes derived from cooperative uranium and non-innocent multielectron redox couples involving charge loaded arenes...."

è -> The sentence is incorrect; indeed, in the reaction of $[\text{Cp}^*2\text{UCl}_2][\text{Cl}]$ with PhN=NPh giving the U(VI) diimido complex $[\text{Cp}^*2\text{U(=NPh)}_2]$ there is no charge loaded arene ligand coordinated to the metal. See *Angew. Chem. Int. Ed.* 1998, 37, No. 7, 959.

RESPONSE: We thank the reviewer for pointing this paper out. It describes oxidation of $\text{Cp}^*2\text{UCl}(\text{NaCl})$ to $\text{Cp}^*2\text{U}(\text{NAd})_2$. That is formally a 3 electron oxidation of U(III) to U(VI), but in one view the former is a U(II) equivalent to give a 4 electron oxidation overall with production of half an equivalent of U(IV) Cp^*2UCl_2 , which is seen experimentally. We have added this reference as it is certainly interesting multi-electron redox chemistry, though it is not oxidative addition.

Reviewer #4 (Remarks to the Author):

The report only considers the crystallographic aspects of the submission.

Four structures are reported, two of which are of isostructural. The structures are well determined and details have been provided in the CIFs concerning the modelling of disorder and solvent treatment through SQUEEZE. No changes to the structures reported are required.

Although documented that this is the cause of the discrepancy in the calculated and reported formula, the standard practice has been changed so the solvent accounted for by Squeeze is no longer given explicitly in the formula.

Otherwise the only issue leading to A and B alerts in the checkcif report is that there are significant residual electron density peaks and troughs around the heavy atom. While it is not unreasonable that there are large peaks adjacent to heavier atoms that are only a small percentage of the total electron density of the U atom, the holes in the difference Fourier maps are larger than expected for 2.pmdeta. It is not suggested that structure is incorrect or action required in this case but in future it may be that it could be improved by re-examination of the diffraction data to see if the crystal was split or twinning is present that has not been resolved. In general F_{obs} is greater than F_{calc} for this refinement and there are many high angle systematic absences with significant intensity which would be consistent with contributions from another component or the data collection or integration parameters being set such that intensity from adjacent reflections was included.

I suggest modifying the text describing the structures of 2.tmeda and 2.pmdeta given on page 7 of the manuscript. The authors highlight how the structure appears to have a very exposed U ion without the stabilising amine but this could be said of many structures if you virtually remove a significant part of the co-ordination sphere of an ion. As shown by the structures of 3 the formation of dimers is also a way to complete coordination. The authors should clarify exactly what they mean to highlight here and change the text accordingly.

RESPONSE: We have reworded the sentence dealing with this discussion point. We have checked for twinning but could not find any.

---End---

REVIEWERS' COMMENTS:

Reviewer #1 (Remarks to the Author):

In this revision, Liddle and co-workers have addressed many of the reviewer comments previously submitted, and have explained why pieces of data, such as ^{13}C NMR and characterization of the products from $2e^-$ RE (reviewer #3), cannot be obtained, though this is unfortunate. I still contend that the calculations presented should be performed in ΔG , even if they are ultimately included in the SI in addition to those in ΔH . The authors claim these calculations were not performed due to potential error and cite an article in ACS Catalysis by Castro et al, which describes coordination-insertion chemistry of olefins. I feel that this is a quite weak argument as the chemistry described in the cited article does not involve any reduction or oxidation at the active metal center. Therefore it is a bit far reaching to claim similar results would be obtained for their work. Because of this, I recommend that revisions are still required before this manuscript may be accepted for publication.

Reviewer #2 (Remarks to the Author):

I am satisfied regarding my earlier comments recommending exploration of alternative trapping agents, and reiterate my belief in the novelty of this contribution.

In reading the response to another reviewer, I am puzzled the author's assertion that earlier examples cited are "multi-electron transfer reactions", but not oxidative additions. I agree that the current submission demonstrates a novel reaction, but do not agree that this is the first bona fide example of oxidative addition. Other examples in uranium chemistry, including those cited (including cleavage of azobenzene), certainly yield oxidized metal centers and new metal-ligand bonds derived from the reduced ligands, and should be considered oxidative addition, even if not all the electron equivalents came from the metal center. Oxidative additions can certainly be more than two-electron; four-electron oxidative addition reactions have been observed in d-transition metal chemistry, (e.g. tungsten chemistry, cf. JACS, 1990, 112, 2298), yet these would not satisfy the perhaps overly restrictive criteria cited by these authors. The novelty of this contribution does not rest on the fact that this is the first oxidative addition (I believe other reactions qualify), nor is it the fact that it might yield interesting catalytic reactions (given that catalytic reduction of azobenzene has already been demonstrated, per reference 49).

I am not even certain it is fair to assert that this is a "two electron oxidative addition followed by a two electron reductive elimination on a single metal ion", as stated in the authors' response. The key mechanistic assertion, based on theory, is that a key intermediate is an initially generated uranium (IV) adduct of a singly-reduced azobenzene (one-electron oxidation) with a second equivalent of the U(III) synthon gives rise to the uranium(V) imido dimer. This is not a simple two-electron oxidative addition at a single metal ion. Further, I agree with the other referee that although the extrusion of azobenzene is indeed noteworthy, there is no evidence of the formation of **2** in this extrusion (despite their admittedly strenuous attempts to characterize the uranium-containing product). This is an assumption.

What is novel is the two-electron oxidation and subsequent extrusion of the initial ligand, yielding net stoichiometric reversibility (even it is not experimentally proven that it is mechanistically simple reversibility). That suggests a fairly subtle electronic balance, which is interesting to explore, and could certainly lead to some very interesting uranium-mediated chemical transformations. This fact merits reporting, without overinterpreting.

Reviewer #3 (Remarks to the Author):

Considering the responses of the authors on the manuscript NCOMMS-17-11819A, I accept the publication

I suggest the authors to re-read carefully their manuscript and the SI and the connection between the two, because to my mind, some errors remain

For example, page 11 of the revised manuscript :

-the values of 24.7 kcal/mol in the text doesnot match with the values of 25.7 kcal mol in the Figure 5

-In addition, always page 11 and to lines later, « ... computed distance of 1.338 angstroem for free azobenzene in the gaz phase » would be (for me) 1.257 angstroem as refered to the Figure S25 in the SI. The values of 1.338 A corresponds to the distance of the coordinated azobenzene in intermediate int-A and not for free azobenzene

Reviewers' comments:

Reviewer #1 (Remarks to the Author):

In this revision, Liddle and co-workers have addressed many of the reviewer comments previously submitted, and have explained why pieces of data, such as ^{13}C NMR and characterization of the products from 2e- RE (reviewer #3), cannot be obtained, though this is unfortunate. I still contend that the calculations presented should be performed in ΔG , even if they are ultimately included in the SI in addition to those in ΔH . The authors claim these calculations were not performed due to potential error and cite an article in ACS Catalysis by Castro et al, which describes coordination-insertion chemistry of olefins. I feel that this is a quite weak argument as the chemistry described in the cited article does not involve any reduction or oxidation at the active metal center. Therefore it is a bit far reaching to claim similar results would be obtained for their work. Because of this, I recommend that revisions are still required before this manuscript may be accepted for publication.

RESPONSE: We have amended figure 8 to provide ΔH and ΔG values and added some text to explain that and why we focus on the former in discussion. We do this because the error made in computing the Gibbs Free Energy is not associated with the type of reaction (redox or coordination-insertion), but rather to the way the entropy is computed within the harmonic approximation, used in all cases. Therefore, the report by Castro et al. about this really does apply here too.

Reviewer #2 (Remarks to the Author):

I am satisfied regarding my earlier comments recommending exploration of alternative trapping agents, and reiterate my belief in the novelty of this contribution.

In reading the response to another reviewer, I am puzzled the author's assertion that earlier examples cited are "multi-electron transfer reactions", but not oxidative additions. I agree that the current submission demonstrates a novel reaction, but do not agree that this is the first bona fide example of oxidative addition. Other examples in uranium chemistry, including those cited (including cleavage of azobenzene), certainly yield oxidized metal centers and new metal-ligand bonds derived from the reduced ligands, and should be considered oxidative addition, even if not all the

electron equivalents came from the metal center. Oxidative additions can certainly be more than two-electron; four-electron oxidative addition reactions have been observed in d-transition metal chemistry, (e.g. tungsten chemistry, cf. JACS, 1990, 112, 2298), yet these would not satisfy the perhaps overly restrictive criteria cited by these authors. The novelty of this contribution does not rest on the fact that this is the first oxidative addition (I believe other reactions qualify), nor is it the fact that it might yield interesting catalytic reactions (given that catalytic reduction of azobenzene has already been demonstrated, per reference 49).

I am not even certain it is fair to assert that this is a "two electron oxidative addition followed by a two electron reductive elimination on a single metal ion", as stated in the authors' response. The key mechanistic assertion, based on theory, is that a key intermediate is an initially generated uranium (IV) adduct of a singly-reduced azobenzene (one-electron oxidation) with a second equivalent of the U(III) synthon gives rise to the uranium(V) imido dimer. This is not a simple two-electron oxidative addition at a single metal ion. Further, I agree with the other referee that although the extrusion of azobenzene is indeed noteworthy, there is no evidence of the formation of 2 in this extrusion (despite their admittedly strenuous attempts to characterize the uranium-containing product). This is an assumption.

What is novel is the two-electron oxidation and subsequent extrusion of the initial ligand, yielding net stoichiometric reversibility (even it is not experimentally proven that it is mechanistically simple reversibility). That suggests a fairly subtle electronic balance, which is interesting to explore, and could certainly lead to some very interesting uranium-mediated chemical transformations. This fact merits reporting, without overinterpreting.

RESPONSE: Oxidative addition has a specific definition where the 2e variant is concerned. We use and teach this to provide a framework of classifications of reactions. Therefore, we cannot pick and choose what to apply and when. If a reaction involves introduction of ligands and redox change that do not satisfy the definition of oxidative addition then they should perhaps be described as multi-electron reactions; indeed some of the spectacular multi-electron oxidations performed by U in recent years have only ever been referred to as multi-electron reactions because they are so different to classical oxidative addition chemistry. In other U examples mentioned, the electrons come from a ligand, usually arene, and not the metal. So they are not oxidative additions because the metal was already oxidized before that reaction occurs; these are ligand redox reactions. The JACS example is interesting. It could be formulated as a 4e oxidation, but it could also be formulated as a 2e oxidation if the carbene is considered as a singlet methylene carbene rather than triplet alkylidene, though we see why the authors of that paper favoured the former designation. Nevertheless, it does not affect the work here because our comparisons within U chemistry hold and as the reviewer mentions the reversibility is notable. Regarding mechanism, a key point about reaction classifications is that they describe the overall reaction. The mechanism, in a sense, is neither here nor there; indeed there are various mechanisms for oxidative addition (concerted, S_N2 , ionic, radical), but although they tread different paths they still reach the same destination.

Reviewer #3 (Remarks to the Author):

Considering the responses of the authors on the manuscript NCOMMS-17-11819A, I accept the publication

I suggest the authors to re-read carefully their manuscript and the SI and the connection between the two, because to my mind, some errors remain

For example, page 11 of the revised manuscript :

-the values of 24.7 kcal/mol in the text does not match with the values of 25.7 kcal mol in the Figure 5

-In addition, always page 11 and to lines later, « ... computed distance of 1.338 angstroem for free azobenzene in the gaz phase » would be (for me) 1.257 angstroem as referred to the Figure S25 in the SI. The values of 1.338 Å corresponds to the distance of the coordinated azobenzene in intermediate int-A and not for free azobenzene.

RESPONSE: Fair points, and we have checked the numbers and corrected a couple of instances of typos.

---End---